# Cancer Predisposition Syndromes and Thyroid Cancer: Keys for a Short Two-Way Street

**DOI:** 10.3390/biomedicines11082143

**Published:** 2023-07-29

**Authors:** Ioana Balinisteanu, Monica-Cristina Panzaru, Lavinia Caba, Maria-Christina Ungureanu, Andreea Florea, Ana Maria Grigore, Eusebiu Vlad Gorduza

**Affiliations:** 1Endocrinology Department, “Grigore T. Popa” University of Medicine and Pharmacy, 700115 Iasi, Romania; dr.balinisteanu.ioana@gmail.com (I.B.); maria.ungureanu@umfiasi.ro (M.-C.U.); 2Endocrinology Department, “Sf. Spiridon” Hospital, 700106 Iasi, Romania; 3Department of Medical Genetics, Faculty of Medicine, “Grigore T. Popa” University of Medicine and Pharmacy, 700115 Iasi, Romania; andreeaflorea97@gmail.com (A.F.); vgord@mail.com (E.V.G.)

**Keywords:** cancer predisposition syndromes, thyroid cancer, heterogeneity

## Abstract

Cancer predisposition syndromes are entities determined especially by germinal pathogenic variants, with most of them autosomal dominantly inherited. The risk of a form of cancer is variable throughout life and affects various organs, including the thyroid. Knowing the heterogeneous clinical picture and the existing genotype–phenotype correlations in some forms of thyroid cancer associated with these syndromes is important for adequate and early management of patients and families. This review synthesizes the current knowledge on genes and proteins involved in cancer predisposition syndromes with thyroid cancer and the phenomena of heterogeneity (locus, allelic, mutational, and clinical).

## 1. Introduction

Thyroid cancer is an important oncological pathology representing approximately 3% of malignancies and 95% of endocrine cancers [1].

From a histological point of view, there are two major types of thyroid cancer: C-cell-derived medullary thyroid carcinoma (5%) and well-differentiated cancers of follicular origin (95%), with the last category being divided into two types: papillary thyroid carcinoma (PTC, 80–90%) and follicular thyroid carcinoma (FTC, 10–15%) [2]. Due to the accumulation of knowledge regarding the origin of the cells, the cyto- and histopathological characteristics, and the molecular and clinical features, in 2022, WHO released a more extensive classification of endocrine and neuroendocrine tumours related to the thyroid gland (the 5th edition). Accordingly, the tumours are categorised into eight groups: 1. Developmental abnormalities (thyroglossal duct cyst and other congenital thyroid abnormalities), 2. Follicular cell-derived neoplasms (benign tumours, low-risk neoplasms, and malignant neoplasms), 3. Thyroid C-cell-derived carcinoma (the medullary thyroid carcinoma), 4. Mixed medullary and follicular cell-derived carcinomas (medullary-follicular and medullary-papillary), 5. Salivary gland-type carcinomas of the thyroid (mucoepidermoid carcinoma of the thyroid and secretory carcinoma of salivary gland type), 6. Thyroid tumours of uncertain histogenesis (sclerosing mucoepidermoid carcinoma with eosinophilia and the cribriform morular thyroid carcinoma, with the last one previously considered a variant of papillary thyroid carcinoma), 7. Thymic within the thyroid (thymoma family, spindle epithelial tumour with thymus-like elements and thymic carcinoma family—intrathyroidal thymic carcinoma), and 8. Embryonal thyroid neoplasms (thyroblastoma). In 2017, WHO recognised only the follicular adenoma as a follicular benign tumour, but in the 2022 classification, there were four types: the thyroid follicular nodular disease, the follicular adenoma pure or with papillary architecture, and the previously called Hürthle-cell adenoma, which is now called oncocytic adenoma. The follicular low-risk neoplasms group includes the non-invasive follicular thyroid neoplasm with papillary-like nuclear features, the thyroid tumours of uncertain malignant potential, and the hyalinizing trabecular tumour. The malignant neoplasms include the classical types (FTC and PTC), the anaplastic carcinoma plus the invasive encapsulated follicular variant papillary carcinoma, the oncocytic carcinoma, and the follicular-derived carcinomas-high-grade (subdivided as differentiated high-grade and poorly differentiated) [3,4,5].

Thyroid cancer can be isolated or be a part of cancer predisposition syndromes.

Cancer predisposition syndromes are diseases characterised by great genetic heterogeneity and variable expressivity. There are two categories of clinical manifestations: predisposition to various forms of cancer and extratumoural manifestations. Sometimes, extratumoural manifestations appear before tumours. Early diagnosis is important for oncological surveillance in these conditions, reducing cancer morbidity and mortality. Most of these syndromes have an autosomal dominant inheritance with a high risk of transmitting the pathogenic variants to the offspring (but are dependent on penetrance). In these conditions, genetic counselling with the identification of people at risk in the family becomes a mandatory objective of case management [6,7].

## 2. Cancer Predisposition Syndromes with Thyroid Cancer

Many cancer predisposition syndromes with thyroid cancers are known. The most important are summarised in Table 1.

Only 0.1% of all thyroid cancer cases could be associated with each of the following cancer predisposition syndromes: Cowden syndrome, Werner syndrome, Carney complex, and familial adenomatous polyposis. For MEN2A and MEN2B, the rate is marginally higher: 0.3% and 0.2% [12].

The age of thyroid cancer diagnosis is different in these syndromes (Figure 1).

The proteins encoded by the genes involved in hereditary syndromes with thyroid cancer intervene in the maintenance of genome integrity, the DNA repair process, miRNA formation and maturation, various signalling pathways, and mitochondrial processes [13].

### 2.1. Familial Adenomatous Polyposis

#### 2.1.1. Clinics and Genetics

Familial adenomatous polyposis (FAP) is characterised by the development of multiple colorectal adenomas, often with progression to colorectal malignancy. FAP1 is an autosomal dominant disease caused by a heterozygous pathogenic variant in the *APC* gene. Penetrance is approximately 100% [14]. FAP1 can present hundreds to thousands of adenomas and variable extracolonic features such as congenital hypertrophy of the retinal pigment epithelium (CHRPE), osteomas, dental abnormalities (teeth agenesis, supernumerary teeth, odontomas), desmoid tumours, adrenal masses, or other associated tumours (thyroid, liver, bile ducts, pancreas, central nervous system). The APC protein has several domains. The domains are an oligomerisation domain, an armadillo repeat-domain, a 15- or 20-residue repeat domain, SAMP repeats domain, a basic domain, and C-terminal domains [15]. Each of these, through the different effectors to which they are linked, intervene in various cellular processes: the regulation of cell adhesion, proliferation and cellular differentiation via the Wnt pathway, the regulation of cell adhesion and migration, the regulation of cell division through the action on chromosomal segregation, and mitotic progression [15,16,17]. The main pathway where the APC protein intervenes is the canonical Wnt pathway, which is ß-catenin dependent. The adenomatous polyposis coli protein has seven ß-catenin binding sites. Each such binding site has a sequence of 20-amino acid repeats (20-AARs). After fixation of ß-catenin to the site, degradation of ß-catenin occurs [18].

Most germline pathogenic variants are in the 5’ half of the *APC* gene. The most frequent types of pathogenic variants are truncating frameshift (67%) and nonsense (28%) [19].

Studies reported an increased risk of thyroid cancer in FAP1 patients with pathogenic variants at the 5′ end of the *APC* gene, proximal to codon 938 or in codon 1061 [16,20,21].

FAP2 is an autosomal recessive disease caused by a biallelic pathogenic variant in the *MUTYH* gene. Patients with FAP2 usually present a smaller number of adenomas (10–100, similar to an attenuated form of FAP1), serrated polyps, and an increased risk of other cancers (duodenum, ovary, bladder, breast, and endometrium). Other manifestations include sebaceous adenomas, thyroid nodules, CHRPE, adrenal lesions, and mandibular cysts [22]. The risk of thyroid malignancy in FAP2 is still unclear, although a few cases of papillary thyroid cancer have been reported [23].

In the cases of FAP3 (caused by biallelic pathogenic variant in *NTHL1*) and FAP4 (caused by biallelic pathogenic variant in *MSH3*), there are limited data that concern thyroid pathology.

#### 2.1.2. Thyroid Cancer

Lately, studies are reporting an increased incidence of thyroid cancer in FAP1 (1–12%) [24], which appears to be due to a greater availability of tests for early detection as well as an increase in ultrasound examination for this particular category of patients. The mean age of diagnosis is 36.5–44 years, with it being earlier in females than males. All studies report an increased incidence of thyroid cancer in female FAP patients (60–80%) [25,26] 160 times greater than in the general population [27]. The familial history of thyroid cancer is unusual, but it can be positive in some cases [26,28].

Usually, the tumours have benign features and small dimensions (median maximum diameter 1.0 cm), while other reports showed an even smaller maximal size between 0.5 and 1.1 cm [25,28], but the multifocality and the fine-needle aspiration cytology can guide the diagnosis [28,29].

As histological types, the most commonly reported is papillary cancer (75–89.9%) [25,26,30], followed by the follicular type (6.1%). Medullary and anaplastic carcinomas are very rare [25].

A rare subtype of thyroid cancer, frequently associated with FAP, is the cribriform morular thyroid carcinoma, and is mostly reported in females younger than 35 years old, with a median age 24.8–26 years [29]. In the study by Park et al., all the patients who presented FAP-associated with cribriform morular thyroid carcinoma had multiple tumours, but all had an excellent response to initial therapy [28].

### 2.2. Cowden Syndrome

#### 2.2.1. Clinics and Genetics

Cowden syndrome (CWS) is an autosomal dominant disorder caused by pathogenic variants in genes involved in the PTEN/PI3K/AKT/mTOR signalling pathway and characterised by multiple hamartomas and cancer predisposition [31]. The PTEN/PI3K/AKT/mTOR signalling pathway is also critical in the development and progression of thyroid cancer [1]. Major clinical criteria for CWS are dysplastic cerebellar gangliocytoma (Lhermitte–Duclos disease), macrocephaly, mucocutaneous lesions (trichilemmomas, acral keratosis, mucocutaneous neuromas, papillomatous papules, penile pigmentation), breast cancer, epithelial thyroid cancer (especially follicular), endometrial carcinoma, and gastrointestinal hamartoma. Minor clinical criteria include benign thyroid abnormalities (multinodular goitre, adenomas), autism, intellectual disability, lipomas, glycogenic acanthosis, uterine leiomyomas, renal cancer, and vascular anomalies [32]. A variable percentage of CWS patients (25–85% depending on selection criteria) have pathogenic variants in *PTEN* (CWS1) [33]. The *PTEN* gene is a tumour suppressor gene. The *PTEN* gene product is part of the tyrosine phosphatase superfamily. The substrate of this phosphatase is lipid phosphatidyl inositol 3,4,5 triphosphate (PIP3). The PTEN protein is an essential antagonist in Akt activation. It has a role in tumour suppression, development, and control of the growth process [18]. Early-onset CWS (with multiorgan involvement) is usually associated with *PTEN* pathogenic missense variants, while late-onset CWS is associated with *PTEN* pathogenic truncating variants. Missense variants are frequently located in the phosphatase domain, a critical region for PTEN enzymatic activity. Truncating variants are usually located in C2 and the phosphatase domain, with a hotspot in exon 5. Studies have reported that cases with benign and malign thyroid pathology, breast cancer, and cutaneous lesions usually presented truncating variants. Hendricks et al. suggested a potential tissue preference for this type of pathogenic variant [34,35]. Up to 30% of cases with CWS phenotype without *PTEN* pathogenic variants have a germline *KLLN* promoter hypermethylation (CWS4). *KLLN* located upstream of *PTEN*, is transcribed in the opposite direction, and both genes share the transcription site and a bidirectional promoter. Patients with *KLLN* epimutation have a higher risk of developing breast, renal, and thyroid cancer [31,36]. Less than 10% of *PTEN*-negative cases have pathogenic variants in genes involved in the PTEN signalling pathway, such as *PIK3CA* (especially in the C2 domain) (CWS5) and *AKT1* (CWS6) (genes, also associated with segmental overgrowth syndrome) [33].

A variant of unknown significance in the *SEC23B* gene has been reported in five affected members of a large family (CWS7) [37]. SEC23B, a component of coat protein complex II (COPII), is involved in ER stress and ribosome biogenesis.

Other cases presented pathogenic variants in succinate dehydrogenase (SDH) complex subunit genes (*SDHB*, *SDHC*, and *SDHD*; generally known as *SDHx*). The SDH complex is a key enzyme involved in Krebs cycle and the electron transport chain. Oncogenic properties are the result of increased levels of reactive oxygen species leading to decreased *P53* gene expression and a resistance to apoptosis [31].

Also, *WWP1* gain-of-function pathogenic variants have been reported in cases with CWS and intestinal oligopolyposis and/or colorectal malignancy. WWP1, an E3 ubiquitin ligase, inhibits PTEN dimerisation [38].

#### 2.2.2. Thyroid Cancer

Approximately 35% of people with CWS will develop thyroid cancer [31,39], with a higher incidence in females (93.7%) than in males (6.3%). The median age is 44 years with a range between 7 and 83.7 years, while 2.9% of the included subjects were paediatric. The histological findings report 55.1% papillary forms, 19.5% papillary–follicular variant, 10% follicular, 1.2% Hürthle cell, 0.3% medullary, 0.3% anaplastic, and 13.7% unknown variants. The follicular type was associated with *PTEN* gene pathogenic variants, which were found to be the most common in the paediatric population, while the papillary variant is associated with *SDHx* and *KLLN* genes alterations [40]. Penetrance of these forms of cancer is approximately 90–95% [14].

### 2.3. Carney Complex Type 1

#### 2.3.1. Clinics and Genetics

Carney complex type 1 is an autosomal dominant disorder characterised by mucocutaneous pigmentary anomalies, myxomas, endocrine abnormalities (overactivity or tumours), and schwannomas. Mucocutaneous anomalies include pale brown to black lentigines (located on the face, conjunctiva, lips, vaginal or penile mucosa) and epithelioid blue nevi. Myxomas are usually located in the heart, skin, and breast. The most frequent endocrine abnormalities are primary pigmented nodular adrenocortical disease (PPNAD), thyroid carcinoma or multiple adenomas, large-cell calcifying Sertoli cell tumours, and growth hormone (GH)-producing adenoma. The protein encoded by *PRKAR1A* is cAMP-dependent protein kinase type I-alpha regulatory subunit. It is a serine/threonine kinase involved in cyclic AMP pathway. It has effects in proliferation and cell differentiation, and also in the apoptosis process [41]. A pathogenic variant of *PRKAR1A* has been detected in more than 70% of cases. The pathogenic variants are located more frequently in exons 3, 7, and 8 with two hotspots: c.491-492delTG and c.709-7del6. Patients with c.491-492delTG frequently presented cardiac myxomas, lentigines, and thyroid tumours, while individuals with c.709-7del6 had PPNAD. Intronic (splice site) pathogenic variants lead to a milder phenotype [42,43]. Somatic pathogenic variants in *PRKAR1A* have been reported in sporadic thyroid carcinomas [44]. Penetrance of this form of cancer is unknown [14].

#### 2.3.2. Thyroid Cancer

Even though it is an uncommon feature which can appear late in the disease’s natural history, approximately 3% of patients with Carney complex have been found to develop thyroid cancer, including both papillary and follicular types [45,46,47].

A more recent study included 26 patients who underwent thyroidectomy, of which 7 presented follicular carcinoma (2 of them multifocal) and 3 papillary carcinoma. Most of the patients were in their middle years (mean age, 46 years; range, 22–59 years). At the time of their identification, the tumours in 4 patients were >3 cm in diameter, with metastases, indicating aggressive types. It is recommended to perform an ultrasonographic examination of the thyroid gland regularly [45].

### 2.4. Werner Syndrome

#### 2.4.1. Clinics and Genetics

Werner syndrome is an autosomal recessive disorder characterised by accelerated aging with onset in the third decade of life and increased cancer susceptibility. Characteristic features include the absence of a pubertal growth spurt, short stature, greying and thinning of scalp hair, hoarse voice, bilateral cataracts, dermatologic abnormalities (scleroderma-like skin atrophy, chronic ankle ulcers, pinched faces), diabetes mellitus, osteoporosis, hypogonadism, and premature atherosclerosis.

The *WRN* gene encodes bifunctional 3’-5’ exonuclease/ATP-dependent helicase WRN. This protein has DNA-helicase activity and also 3’- > 5’ exonuclease activity [10]. It is mainly linked to replication forks and Holliday junctions. It has a role in DNA replication and transcription, and recombination and repair (homologous and non-homologous recombination, base excision repair pathways) [48,49].

The majority of cases present biallelic *WRN* truncating pathogenic variants [50]. *WRN* founder pathogenic variants have been reported in some populations: c.3139-1G > C in Japanese and c.2089-3024A > G in Sardinian people [51]. The most common neoplasms in WS are sarcomas of mesenchymal origin, melanomas, and thyroid carcinomas [52]. The localisation of *WRN* pathogenic variants seems to correlate with the type of thyroid cancer: papillary thyroid carcinoma has been reported in cases with an N-terminal variant, and follicular thyroid in patients with a C-terminal variant [53].

#### 2.4.2. Thyroid Cancer

Werner syndrome has a higher prevalence in Japan. The incidence of thyroid cancer in Japanese patients with this pathology is also 8.9-fold higher compared to the general population [52]. The average age for developing this neoplasia is 39 years, with a significant predominance in males. The thyroid follicular carcinoma is the most common form of neoplasia cited in Werner syndrome. Other reported histological types are papillary and occasionally anaplastic thyroid cancer [53,54].

### 2.5. McCune–Albright Syndrome

#### 2.5.1. Clinics and Genetics

McCune–Albright syndrome (MAS), caused by a somatic activating pathogenic variant in the *GNAS* gene, is defined by the triad of polyostotic fibrous dysplasia (FD), irregular “café au lait” skin macules, and endocrinopathies. Endocrine involvement includes gonadotropin-independent precocious puberty, thyroid abnormalities (with or without hyperthyroidism), growth hormone excess, Cushing syndrome, and renal phosphate wasting. The phenotype is variable and is determined by the degree of mosaicism and the location of tissues in which the pathogenic variant is present.

The *GNAS* gene is a gene with complex genomic imprinting and bicistronic expression. Thus, there are three categories of proteins: neuroendocrine secretory protein 55 (maternally derived), guanine nucleotide-binding protein G(s) subunit alpha isoforms XLas (paternally derived), and guanine nucleotide-binding protein G(s) subunit alpha isoforms short (Gsα). The first two are expressed exclusively in neuroendocrine tissues [10,55,56]. Gsα is ubiquitously expressed, interfering with the cAMP pathway and having GTPase activity. It acts as a signal molecule for hormones, neurotransmitters, and paracrine and autocrine factors [56].

More than 95% of patients have a pathogenic variant in codon 201 (p.Arg201His, p.Arg201Cys), leading to the substitution of arginine, a crucial element for the GTPase activity. Another less commonly pathogenic variant concerns codon 227—p.Gln227Leu. Malignant transformation of FD has been reported, especially in patients exposed to radiation. Also, radioablation is not recommended for hyperthyroidism, because of the increased risk of cancer in the remaining gland. Other tumours detected in MAS patients include testicular, pancreatic (intraductal papillary mucinous neoplasm), breast, prostate, and liver (hepatoblastoma) [57,58,59].

#### 2.5.2. Thyroid Cancer

Even though hyperthyroidism is common in this syndrome, thyroid cancer is rare. There are a few cases reported, including an incidentally found papillary carcinoma and a rare variant—clear cell carcinoma. Fine needle aspiration biopsy (FNAB) may not be able to distinguish between follicular thyroid adenoma and follicular carcinoma, whence the need for regular ultra-sonographic surveillance and elastography [60].

### 2.6. DICER1 Syndrome

#### 2.6.1. Clinics and Genetics

DICER1 syndrome is a tumour predisposition disorder caused by heterozygous pathogenic variants in the *DICER1* gene. The most common manifestations include lung cysts that can progress to pleuropulmonary blastoma, multinodular goitre and differentiated thyroid cancer, macrocephaly, ovarian tumours (Sertoli–Leydig cell tumour, embryonal rhabdomyosarcoma), and cystic nephroma. Less frequently detected tumours are central nervous system tumours (sarcoma, pituitary blastoma, pineoblastoma), ciliary body medulloepithelioma, nasal chondromesenchymal hamartoma, embryonal rhabdomyosarcoma, cystic hepatic lesions, Wilms tumour, and presacral malignant teratoid tumour.

The *DICER1* gene encodes a ribonuclease (RNase III) named Dicer that intervenes in the process of pre-miRNA or double-stranded RNA cleavage and miRNA formation [61]. MicroRNAs are involved in the post-transcriptional regulation of 30% of protein-coding genes. Canonical biogenesis of miRNAs involves the formation of primary miRNAs (pri-miRNAs) and subsequent processing of miRNAs (pre-miRNAs) and then mature miRNAs [62]. Dicer endonuclease acts in the cytoplasm in the formation of mature duplex miRNAs. One of the 5p or 3p strands is loaded into Argonaute proteins and in this way forms the miRNA-induced silencing complex (miRISC). miRISC binds to the target messenger RNA and subsequently inhibits translation [62,63].

Most cases have a germline *DICER1* heterozygous pathogenic variant, leading to an increased risk of developing tumours. Tumour tissue sequencing usually detects biallelic, compound pathogenic variants of *DICER1*, while a second pathogenic variant is probably acquired during tumourigenesis, consistent with Knudson’s two-hit hypothesis. Usually, the germline variant is a truncating variant (nonsense or frameshift) that interrupts the open reading frame before the end of the essential RNase IIIb domain, resulting in a loss of protein function. The somatic variant is a missense located in some hotspots (E1705, D1709, G1809, D1810, and E1813) and affects the metal binding sites of the RNase IIIb domain [61,63]. Approximately 10% of cases have no germline *DICER1* pathogenic variant but have a somatic mosaic for a *DICER1* tumour predisposition variant (detected in many tissues, not only in tumours). Cases with mosaic loss-of-function variants (LOF) have a milder phenotype than patients with germline LOF variants, probably due to the reduced number of cells at risk for a second hit. Surprisingly, more severe phenotype (early onset, multiple foci of disease developed simultaneously or successively, involving many tissues/organs) is present in patients with mosaic missense variants on one allele and an acquired LOF in the second allele. Brenneman et al. suggested that the probability of a LOF variant is much higher than for a missense variant, so those cells with a pre-existing missense variant are at a high risk of acquiring a subsequent LOF mutation. Biallelic LOF variants in the *DICER1* gene have not been reported, suggesting that the complete loss of the *DICER1* gene is not viable [64].

#### 2.6.2. Thyroid Cancer

Khanet al. performed a study comparing the number of thyroid cancers listed in the National Institute of Health Clinical Centre DICER1 database with matched data from the Surveillance, Epidemiology, and End Results Program. They reported an increased risk of 16- to 24-fold among the *DICER1* pathogenic variants carriers. They found well-differentiated histological types, both papillary and follicular. The tumours are typically encapsulated, without extrathyroidal extension, and are not more aggressive or less responsive to treatment. Poorly differentiated thyroid cancer is rare. There is a correlation between the occurrence of thyroid cancer and the chemotherapy received for pleuropulmonary blastoma, but there are also cases who did not undergo chemo- or radiotherapy [65,66].

There are other cases described in the literature, especially among the paediatric population, in association with pleuropulmonary blastoma and Sertoli–Leydig cell tumours [67,68,69]. There is also evidence that paediatric poorly differentiated thyroid carcinoma is a manifestation of DICER1 Syndrome [70].

### 2.7. Peutz–Jeghers Syndrome

#### 2.7.1. Clinics and Genetics

Peutz–Jeghers syndrome (PJS) is characterised by gastrointestinal hamartomatous polyposis, mucocutaneous pigmentation, and an increased lifetime risk to develop malignancies. PJS polyps are usually multilobulated; can occur at any location in the gastrointestinal (GI) tract, but frequently in the small intestine; can associate complications (intussusception, bleeding, rectal prolapse); and may develop malignant changes. Mucocutaneous pigmentation is represented by dark blue to dark brown macules around the mouth, eyes, nostrils, on fingers, buccal mucosa, and in the perianal area. Patients with PJS have a predisposition for colorectal, gastric, pancreatic, breast, gonadal (sex cord tumours with annular tubules, Sertoli cell tumour), small bowel, and uterine neoplasms [71]. Few cases of differentiated thyroid cancer have been reported. PJS is caused by a germline heterozygous pathogenic variant in the *STK11* gene. Penetrance of this form is approximately 95–100% [14]. Serine/threonine-protein kinase STK11, encoded by the *STK11* gene, is a kinase with a role in DNA repair, cellular metabolism, and cell polarity [72]. In certain stress conditions (e.g., low nutrients), AMPK is activated and thus catabolic processes increase and anabolism decreases. Serine/threonine-protein kinase STK11 is also involved in tumourigenesis in the mammalian target of the rapamycin (mTOR) signalling pathway, adipose tissue metabolism, DNA repair, and neuronal polarity [72,73,74]. Truncating pathogenic variants are associated with a severe phenotype (early onset, more GI surgical interventions, and higher risk for tumours). Patients with pathogenic variants in exon 7, affecting protein kinase domain XI, have a higher incidence of dysplastic polyps [75].

#### 2.7.2. Thyroid Cancer

Thyroid carcinoma’s incidence in Peutz–Jeghers syndrome is low, but it seems that it appears at a young age (30 years), whence the need for close surveillance [76,77]. A recent study found a total of 2 thyroid cancer in 96 patients, both papillary. After gastrointestinal, breast, and cervical cancers, thyroid cancers were discovered to be the most prevalent, with a median age at diagnosis of 41.4 years [78].

### 2.8. Ataxia-Telangiectasia

#### 2.8.1. Clinics and Genetics

Ataxia-telangiectasia (A-T) is an autosomal recessive disorder defined by progressive cerebellar ataxia, oculocutaneous telangiectasia, immunodeficiency with recurrent infections, and predisposition to malignancy. Elevated serum alfa fetoprotein and chromosomal instability (spontaneous and induced chromosomal breaks and rearrangements frequently involving chromosomes 7 and 14) are other suggestive findings for A-T. Patients with A-T have an increased incidence of neoplasms, especially lymphoma and leukaemia, but also breast, liver, gastric, and oesophageal malignancies [79]. Patients with A-T and thyroid cancer have been reported [80]. Cells from patients with A-T have an increased sensitivity to ionizing radiation, so standard radiotherapy/radiomimetic drugs to treat cancer should be avoided/used with caution.

The *ATM* gene product is a serine-protein kinase ATM—that is, part of the family of phosphoinositide 3-kinase (PI3K)-related kinases. The main role is in DNA repair. DNA damage can occur during replication, oxidative stress, or the action of endogenous or exogenous factors [81]. Serine-protein kinase ATM contains S/T-Q motifs (serine or threonine followed by glutamine). DNA double strand breaks induce autophosphorylation in multiple sites where serine is present. In this way, the kinase becomes active and the DNA damage response process begins [81]. The protein also intervenes in the regulation of the cell cycle and autophagy (by phosphorylating cellular tumour antigen p53, serine/threonine-protein kinase Chk2, E3 ubiquitin-protein ligase Mdm2), telomere processing, and metabolic regulation [81].

The majority of patients have biallelic truncating pathogenic variants leading to no detectable ATM protein and a classic phenotype. Individuals homozygous or compound heterozygous for certain pathogenic variants (c.3576G > A, c.8147T > C, c.5762-1050A > G) have a residual ATM activity (<15%) and a milder phenotype (later onset and slower progression) [82].

#### 2.8.2. Thyroid Cancer

There is no clear evidence of a higher risk of thyroid cancer in this syndrome, nor management or screening recommendations. The few cases reported seem to appear at an early age (9.3–35.8 years), with both papillary and follicular types having being identified, usually in female subjects [83].

### 2.9. Pendred Syndrome

#### 2.9.1. Clinics and Genetics

Pendred syndrome (PDS) is characterised by a triad of features: sensorineural hearing impairment, temporal bone abnormalities (bilateral enlarged vestibular aqueduct with or without cochlear hypoplasia), and goitre (usually with normal thyroid function). The majority of patients have biallelic pathogenic variants in the *SLC26A4* gene [84]. Pendrin has the role of iodine transporter. It is located in the apical membrane of thyrocytes. Pendrin is responsible for the active transport and accumulation of iodides inside the follicular colloid, which is crucial for the synthesis of thyroid hormones [1].

Digenic inheritance has been reported in rare cases: a heterozygous pathogenic variant in the *SLC26A4* gene and a heterozygous pathogenic variant in either the *FOXI1* or *KCNJ10* genes. FOXI1 binds to a Slc4a9 promoter element and activates transcription. The number of *SLC26A4* pathogenic variants is correlated with the severity of the phenotype. The absence of pendrin in thyroid cells leads to atrophic and hyperplastic changes and the development of follicular nodular disease and, sometimes, tumourigenesis [85,86,87].

#### 2.9.2. Thyroid Cancer

Subjects with Pendred syndrome carry a 1% risk of developing thyroid cancer, potentially linked to prolonged TSH elevation and iodine deficiency. Females are more affected (F/M  =  9:2). The growth pattern is clearly follicular as the histology findings report just follicular and follicular variant of papillary carcinomas [88,89].

### 2.10. Li–Fraumeni Syndrome

#### 2.10.1. Clinics and Genetics

Li–Fraumeni syndrome (LFS) is an autosomal dominant cancer predisposition syndrome characterised by the early onset of multiple neoplasms, and familial aggregation. LFS is associated with a wide spectrum of malignancies, with the most common being soft-tissue sarcomas, osteosarcomas, breast cancer, brain tumours, and adrenal cortical carcinomas. Other malignancies reported in LFS patients include leukaemia, lymphoma, melanoma, gastrointestinal tumours, and cancers of the pancreas, lung, gonads, prostate, endometrium, kidney, and thyroid. The majority of LFS cases have a germline heterozygous pathogenic variant in the *TP53* gene. Penetrance is approximately 90–95% [14]. The *TP53* gene is a tumour suppressor gene [10]. It is the gene that has the most frequent pathogenic variants in human cancers (it suffers somatic pathogenic variants in over 50% of human cancers). Cellular tumour antigen p53 has in its structure: an amino-terminal transactivation domain with two subdomains TAD1 and TAD2, a proline-rich region, a DNA-binding domain and tetramerisation domains, and a carboxyl terminus [90]. Most pathogenic variants are in the region that encodes the DNA-binding domain. Cellular tumour antigen 53 has a role in the control of the cell cycle with the triggering of cell cycle arrest, the triggering of apoptosis, and the repair of DNA lesions as a result of the action of cellular stress [91].

Many pathogenic variants are located in the DNA binding domain in some hotspots: p.Arg175His, p.Gly245Asp, p.Gly245Ser, p.Arg248Gln, p.Arg248Trp, p.Arg273Cys, p.Arg273His, and p.Arg282Trp. Pathogenic variants in these hotspots are associated with reduced transcriptional activity of TP53 and usually have a poor prognosis, but genotype–phenotype correlations are challenging. The type, location of pathogenic variants, and effect on TP53 protein activity seem to correlate with the penetrance of the disease and the prognosis (risk to develop other tumours, type of tumour). Certain missense pathogenic variants have a dominant negative effect and are associated with early onset and high penetrance. Truncating pathogenic variants leading to loss of function have been reported in cases with adult cancers and low penetrance [92]. Some pathogenic variants seem to correlate with a specific type of tumour: the Brazilian founder variant p.Arg337His has been reported especially in cases with adrenocortical tumours [93]. Several genetic modifiers have been suggested to influence the phenotypic variability in LFS: *TP53* p.Arg72 polymorphism, *MDM2* c.14 + 309T > G variant, and microRNA R-605 [92,94]. Reports about the development of other tumours after radiotherapy led to caution/avoidance of radiation therapy and genotoxic chemotherapies where possible [95]. Few families with the LFS phenotype and heterozygous pathogenic variant in the *CHEK2* gene have been reported (LFS2) [96].

#### 2.10.2. Thyroid Cancer

Thyroid cancer occurs in 0.9% of the LFS patients with classic DNA-binding domain pathogenic variants but Formiga et al. reported a higher prevalence (10.9%) in Brazilian *TP53* p.Arg337His carriers [97]. The median age at diagnosis is 44 years and 9% of the patients have a family history of thyroid carcinoma. Histologically, the tumours are classified as pure papillary, with only a few presenting the follicular variant of papillary carcinoma [98].

### 2.11. Multiple Endocrine Neoplasia Type 2

#### 2.11.1. Clinics and Genetics

Gain-of-function *RET* pathogenic variants are potent oncogenic drivers and are associated with multiple endocrine neoplasia type 2 (MEN2) as well as sporadic medullary thyroid carcinoma (MTC). Penetrance is approximately 70–100% [14]. The *RET* gene is a proto-oncogene. The protein encoded by the *RET* gene has three domains: extracellular (with four cadherin-like regions), transmembrane (a cysteine-rich region), and cytoplasmic. The last domain has a tyrosine kinase activity [99]. RET receptor tyrosine kinase plays a major role in cell growth and proliferation, and differentiation [99,100].

MEN2 is classified into three subtypes based on clinical features—MEN2A, MEN2B, and FMTC (which may be a variant of MEN2A). The most common subtype, MEN2A, is characterised by two or more specific endocrine tumours (MTC, pheochromocytoma, or parathyroid adenoma/hyperplasia). Some patients with MEN2A present pruritic cutaneous lichen amyloidosis in the upper back’s scapular region or Hirschsprung disease. FMTC is diagnosed in families with four or more cases of MTC without other manifestations of MEN2A, but the distinction between FMTC and MEN2A is challenging, and in small kindreds, there is the possibility of not identifying MEN2A and the risk of pheochromocytoma. Currently, FMTC is considered a clinical variant of MEN2A with reduced penetrance of pheochromocytoma and hyperparathyroidism. Also, MTC has a later onset in FMTC. MEN2B, the most aggressive subtype of MEN, is characterised by early-onset MTC, pheochromocytoma, and suggestive extra-endocrine features: distinctive faces (neuromas of the lips and tongue, prominent lips), ophthalmologic abnormalities (alacrima, thickened and everted eyelids, mild ptosis, and medullated corneal nerve fibres), skeletal anomalies (marfanoid habitus, pectus excavatum, scoliosis, pes cavus, joint laxity), and ganglioneuromatosis of the gastrointestinal tract [101].

The main pathogenic hotspots in MEN2A are located in the *RET* cysteine-rich domain- exon 10 (codons 609, 611, 618, or 620) and exon 11 (codons 630 and 634). Codon 634 pathogenic variants are associated with a higher risk of developing pheochromocytoma and pruritic cutaneous lichen amyloidosis [101,102,103,104]. Cases with MEN2A and Hirschprung disease usually have pathogenic variants in exon 10. Hirschprung disease is caused by loss-of-function pathogenic variants, so the association with MEN2A is surprising. Possible explanations are either a double function (‘‘Janus effect’’) of these variants—a gain-of-function effect that triggers the proliferation of the thyroids C-cells and adrenal cells and a reduced level of RET at the plasma membrane leading to anomalies in the neural crest cell migration in the colon, and either the abnormal level of RET affects the balance proliferation/differentiation and leads to excessive proliferation and decreased migration [105,106]. Pathogenic variants initially associated only with FMTC have been also reported in MEN2A. These variants result in substitutions of cysteines or other amino acids in intra or extracellular domains – codons 533, 634 (especially C634Y but not C634R) 768, 790, 804, and 891 [104]. MEN2B is associated with pathogenic variants in *RET* kinase domain exons 16 (M918T) and 15 (A883F). Somatic *RET* pathogenic variants and *RET* rearrangements have been also reported in sporadic MTC, papillary thyroid carcinoma, and multiple neoplasms [107]. *RET* variants are key elements for personalised management.

#### 2.11.2. Thyroid Cancer

The prevalence of MEN2A is 13–24 per million and MEN2B is 1–2 per million. The incidence ranges between 8–28 (MEN2A) and 1–3 (MEN2B) per million live births per year [108]. The MTC is the most prevalent and usually the first MEN2 manifestation, with 100% penetrance. It is a neuroendocrine tumour derived from calcitonin-producing parafollicular C-cells and emerges at a young age, with the maximum incidence in the third decade of life. Biologically, the high serum calcitonin levels lead to the diagnosis. Clinically, the tumour presents as a solitary nodule and/or cervical lymph adenopathy, but, histologically, it is usually multicentric and bilateral. MEN2B tends to be more aggressive than MEN2A, a fact sustained by the different pathogenic variants. The treatment is also aggressive, consisting of a total thyroidectomy with lymph node dissection; it should be performed prophylactic starting at one year of age in MEN2B and at age 5 in MEN2A, if the pathogenic variants are high-risk [109].

## 3. Conclusions

Genetic heterogeneity is important in the diagnosis and proper management of syndromes with predisposition to cancer.

Even though a small part of overall thyroid cancers is associated with cancer predisposition syndromes, it is extremely important to identify the pleiotropic manifestations in these syndromes, in order to obtain an earlier diagnosis and to effect better prevention. Another important tool for this goal is a detailed personal and family history. Genetic testing is guided by essential information from clinical assessment and pedigree analysis. Identification of the pathogenic variants is a key element for the familial screening and for patients’ follow-up (other tumours). The time of appearance of thyroid cancer and its particularities (mainly histological) can lead to a precise diagnosis, especially in the case of variable expressiveness. Another benefit of early diagnosis of these syndromes is the possibility to prescribe an adequate treatment.

## Figures and Tables

**Figure 1 biomedicines-11-02143-f001:**
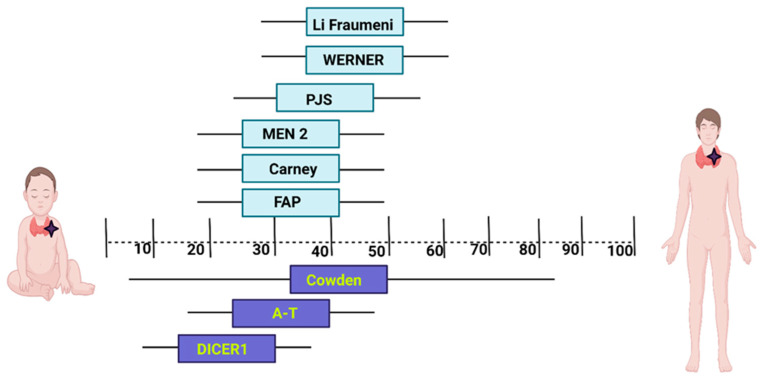
The mean age of thyroid cancer diagnosis in most frequent cancer predisposition syndromes. Created with BioRender.com available at: https://www.biorender.com/ (accesed on 27 June 2023). PJS: Peutz–Jeghers syndrome; MEN2—Multiple endocrine neoplasia type 2; FAP: Familial adenomatous polyposis; A-T: Ataxia-telangiectasia.

**Table 1 biomedicines-11-02143-t001:** Genes and proteins implied in syndromic thyroid cancer [8,9,10,11].

Syndrome	Approved/Alias (Previous), Gene Symbol	Approved Gene Name	Location	Approved Protein Name	Inheritance	OMIM/ Observations
Familial adenomatous polyposis	*APC*/ *DP2*, *DP3*, *DP2.5*, *PPP1R46*	APC regulator of WNT signalling pathway	5q22.2	Adenomatous polyposis coli protein	AD	175100/ FAP1
	*MUTYH/MYH*	Mut Y DNA glycosylase	1p34.1	Adenine DNA glycosylase	AR	608456/ FAP2
	*NTHL1/NTH1, OCTS3*	nth like DNA glycosylase 1	16p13.3	Endonuclease III-like protein 1	AR	616415/FAP3
	*MSH3/DUP*, *MRP1*	Mut S homolog 3	5q14.1	DNA mismatch repair protein Msh3	AR	617100/ FAP4
Cowden syndrome	*PTEN/MMAC1*, *TEP1*, *PTEN1*, (*BZS*, *MHAM*)	phosphatase and tensin homolog	10q23.31	Phosphatidylinositol 3,4,5-trisphosphate 3-phosphatase and dual-specificity protein phosphatase PTEN	AD	158350/ CWS1
	*AKT1/RAC*, *PKB*, *PRKBA*, *AKT*, *RAC-alpha*	AKT serine/threonine kinase 1	14q32.33	RAC-alpha serine/threonine-protein kinase	AD	615109/ CWS6
	*PIK3CA/PI3K*	phosphatidylinositol-4,5-bisphosphate 3-kinase catalytic subunit alpha	3q26.32	Phosphatidylinositol 4,5-bisphosphate 3-kinase catalytic subunit alpha isoform	AD	615108/ CWS5
	*SEC23B/CDA-II*, *CDAII*, *HEMPAS*, (*CDAN2*)	SEC23 homolog B, COPII coat complex component	20p11.23	Protein transport protein Sec23B	AD	616858/ CWS7
	*KLLN*/ *killin*	killin, p53 regulated DNA replication inhibitor	10q23	Killin	AD	615107/ CWS4
Carney complex, type 1	*PRKAR1A*/ *CNC1*, (*PRKAR1*, *TSE1*)	protein kinase cAMP-dependent type I regulatory subunit alpha	17q24.2	cAMP-dependent protein kinase type I-alpha regulatory subunit	AD	160980
Werner syndrome	*WRN/RECQL2*, *RECQ3*	WRN RecQ-like helicase	8p12	Bifunctional 3’-5’ exonuclease/ATP-dependent helicase WRN	AR	277700
McCune–Albright syndrome	*GNAS/NESP55*, *NESP*, *GNASXL*, *GPSA*, *SCG6*, *SgVI*, (*GNAS1*)	GNAS complex locus	20q13.32	Guanine nucleotide-binding protein G(s) subunit alpha isoforms short		174800
DICER1 syndrome	*DICER1/Dicer*, *KIAA0928*, *K12H4.8-LIKE*, *HERNA*, (*MNG1*)	dicer 1, ribonuclease III	14q32.13	Endoribonuclease Dicer	AD	601200, 138800
Peutz–Jeghers syndrome	*STK11/PJS*, *LKB1*	serine/threonine kinase 11	19p13.3	Serine/threonine-protein kinase STK11	AD	175200
Ataxia- telangiectasia	*ATM/TEL1*, *TELO1*, (*ATA*, *ATDC*, *ATC*, *ATD*)	ATM serine/threonine kinase	11q22.3	Serine-protein kinase ATM	AR	208900
Pendred syndrome	*SLC26A4/PDS*, (*DFNB4*)	solute carrier family 26 member 4	7q22.3	Pendrin	AR	274600
Li Fraumeni syndrome	*TP53*/ *p53*, *LFS1*	tumour protein p53	17p13.1	Cellular tumour antigen p53	AD	151623/ LFS
	*CHEK2/CDS1*, *CHK2*, *HuCds1*, *PP1425*, *bA444G*, (*RAD53*)	checkpoint kinase 2	22q12.1	Serine/threonine-protein kinase Chk2	AD	609265/ LFS2
MEN II	*RET*/ *PTC*, *CDHF12*, *RET51*, *CDHR16* (*HSCR1*, *MEN2A*, *MTC1*, *MEN2B*)	ret proto-oncogene	10q11.21	Proto-oncogene tyrosine-protein kinase receptor Ret	AD	162300, 171400

AD—autosomal dominant; AR—autosomal recessive.

## Data Availability

Not applicable.

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
