# Peer review of "Cancer Predisposition Syndromes and Thyroid Cancer: Keys for a Short Two-Way Street"

_biomedicines, 2023, doi:10.3390/biomedicines11082143_

Round 1
Reviewer 1 Report
The manuscript by Balinisteanu and co-workers focuses on advances in the understanding of the cancer predisposition syndromes associated with thyroid cancer. The review article is interesting and well-written. I would suggest some changes, that are listed below.
[1.] Introduction (lines 26-34): I would suggest introducing at least one reference in this section.
[2.] I would suggest that the classification of thyroid cancers be better described in lines 46-49. At the beginning of the manuscript, only papillary, follicular, and medullary thyroid carcinomas were mentioned, although the Authors then refer to other thyroid cancers (e.g., anaplastic or cribriform–morular variant of papillary thyroid carcinoma (CMV–PTC)). It is also worth mentioning the new histologic classification of thyroid neoplasms released in 2022 by the World Health Organization (as described in Endocr Relat Cancer. 2022 Dec 22;30(2):e220293).
[3.] The Authors mentioned that Cowden syndrome is associated with genetic aberrations of the PI3K/AKT/mTOR pathway. It is worth mentioning that this pathway is also critical in in the development and progression of thyroid cancer (as described in Int J Mol Sci. 2021 Oct 31;22(21):11829).
[4.] Describing the Pendred syndrome, the Authors mentioned pendrin protein encoded by the SLC26A4 gene. It is worth mentioning here that pendrin is considered an iodine transporter located in the apical membrane of thyrocytes. It is responsible for the active transport and accumulation of iodides inside the follicular colloid, which is crucial for the synthesis of thyroid hormones.
[5.] Line 250: Please explain FNAB abbreviation.
The manuscript by Balinisteanu and co-workers focuses on advances in the understanding of the cancer predisposition syndromes associated with thyroid cancer. The review article is interesting and well-written. I would suggest some changes, that are listed below.
[1.] Introduction (lines 26-34): I would suggest introducing at least one reference in this section.
[2.] I would suggest that the classification of thyroid cancers be better described in lines 46-49. At the beginning of the manuscript, only papillary, follicular, and medullary thyroid carcinomas were mentioned, although the Authors then refer to other thyroid cancers (e.g., anaplastic or cribriform–morular variant of papillary thyroid carcinoma (CMV–PTC)). It is also worth mentioning the new histologic classification of thyroid neoplasms released in 2022 by the World Health Organization (as described in Endocr Relat Cancer. 2022 Dec 22;30(2):e220293).
[3.] The Authors mentioned that Cowden syndrome is associated with genetic aberrations of the PI3K/AKT/mTOR pathway. It is worth mentioning that this pathway is also critical in in the development and progression of thyroid cancer (as described in Int J Mol Sci. 2021 Oct 31;22(21):11829).
[4.] Describing the Pendred syndrome, the Authors mentioned pendrin protein encoded by the SLC26A4 gene. It is worth mentioning here that pendrin is considered an iodine transporter located in the apical membrane of thyrocytes. It is responsible for the active transport and accumulation of iodides inside the follicular colloid, which is crucial for the synthesis of thyroid hormones.
[5.] Line 250: Please explain FNAB abbreviation.
Author Response
Thank you very much for your comments and constructive observations.
[1.] Introduction (lines 26-34): I would suggest introducing at least one reference in this section.
Answer: We introduced two references.
[2.] I would suggest that the classification of thyroid cancers be better described in lines 46-49. At the beginning of the manuscript, only papillary, follicular, and medullary thyroid carcinomas were mentioned, although the Authors then refer toother thyroid cancers (e.g., anaplastic or cribriform–morular variant of papillary thyroid carcinoma (CMV–PTC)). It is also worth mentioning the new histologic classification of thyroid neoplasms released in 2022 by the World Health Organization(as described in Endocr Relat Cancer. 2022 Dec22;30(2):e220293).
Answer: We described the classification of thyroid cancer using the new histologic classification of thyroid neoplasms released in 2022 by the World Health Organization
[3.] The Authors mentioned that Cowden syndrome is associated with genetic aberrations of the PI3K/AKT/mTOR pathway. It is worth mentioning that this pathway is also critical in in the development and progression of thyroid cancer (as described in Int J Mol Sci. 2021 Oct 31;22(21):11829).
Answer: We added the information
[4.] Describing the Pendred syndrome, the Authors mentionedpendrin protein encoded by the SLC26A4 gene. It is worthmentioning here that pendrin is considered an iodine transporterlocated in the apical membrane of thyrocytes. It is responsiblefor the active transport and accumulation of iodides inside thefollicular colloid, which is crucial for the synthesis of thyroidhormones.
Answer: We added the information
[5.] Line 250: Please explain FNAB abbreviation.
Answer: We explained
Reviewer 2 Report
Balinisteanu and colleagues conducted a comprehensive review on the association of thyroid cancer and 11 cancer predisposition syndromes: familial adenomatous polyposis, Cowden syndrome, Carney complex, Werner syndrome, McCune-Albright syndrome, DICER1 syndrome, Peutz-Jeghers syndrome, ataxia-telangiectasia, pendred syndrome, Li-Fraumeni syndrome, and multiple endocrine neoplasia type 2. Overall, the manuscript is clearly written with some updated pathogenetic information. I have a few minor suggestions:
(1) It should be noted that the 'cribriform-morular variant of papillary thyroid carcinoma' has now been renamed to 'cribriform morular thyroid carcinoma' and has been attributed to thyroid tumors of uncertain histogenesis. (Baloch et al. Overview of the 2022 WHO Classification of Thyroid Neoplasms. Endocr Pathol 2022; 33: 27-63. doi: 10.1007/s12022-022-09707-3.)
(2) Line 473: a reference was not included in the reference list. (Mathiesen et al. Multiple endocrine neoplasia type 2: A review. Semin Cancer Biol 2022;79:163-179. doi: 10.1016/j.semcancer.2021.03.035.)
(3) Reference #101: format error. (Yasir et al. Multiple Endocrine Neoplasias Type 2. 2022 Aug 22. In: StatPearls [Internet]. Treasure Island (FL): StatPearls Publishing; 2023 Jan.)
Author Response
Thank you very much for your comments and constructive observations.
- It should be noted that the 'cribriform-morular variant of papillary thyroid carcinoma' has now been renamed to 'cribriform morular thyroid carcinoma' and has been attributed to thyroid tumors of uncertain histogenesis. (Baloch et al. Overview of the2022 WHO Classification of Thyroid Neoplasms. Endocr Pathol2022; 33: 27-63. doi: 10.1007/s12022-022-09707-3.)
Answer: We replaced in our manuscript with the histologic classification of thyroid neoplasms released in 2022 by the World Health Organization
- Line 473: a reference was not included in the reference list.(Mathiesen et al. Multiple endocrine neoplasia type 2: A review.Semin Cancer Biol 2022;79:163-179. doi:10.1016/j.semcancer.2021.03.035.)
Answer: We introduced the reference
- Reference #101: format error. (Yasir et al. Multiple EndocrineNeoplasias Type 2. 2022 Aug 22. In: StatPearls [Internet].Treasure Island (FL): StatPearls Publishing; 2023 Jan.)
Answer: We introduced the correct format reference according to mdpi style
Reviewer 3 Report
The authors presented an interesting review about cancer predisposition syndromes in thyroid cancer. This review synthesizes current knowledge on genes and proteins involved in cancer predisposition syndromes with thyroid cancer and the phenomena of heterogeneity (locus, allelic, mutational, and clinical). Authors conclude that the time of appearance of thyroid cancer and its particularities can lead to a precise diagnosis, or sooner diagnostics. Moreover, the presented knowledge could be a practical suggestion for oncologists.
However, I have a few major issues that have to be solved or explained:
- I’m wondering if the title is not too sophisticated, what is "second way or second direction" in the street?
- I’m not sure about the order of the syndromes; is it a frequency in Thyroid cancer or the frequency of the syndrome in the population?
- I think that both the introduction and conclusion could be more developed, especially the conclusion in the context of practical patients diagnostics… In the introduction, authors could add some more details about
- I’m wondering if Table 1 could have an additional column with frequency in thyroid cancer, maybe just percentage or "rare"?
-
Please check the correctness of all citation i.e 21, 82 (not cancer), 91 perhaps some citation should be split because they do correspond to the text and are confusing.
-
Please check the correctness of all citation i.e 21, 82 (not cancer), 91 perhaps some citation should be split because they do not correspond to the text and thus are confusing.
- In line 473 problem with the citation format?
A strong point that is crucial for my final recommendation is that I could not find any similar article published recently that would cover the same topic or idea.
- Please carefully check “the” and “a”, syntax, and gramma i.e. Lines: 101 others report (s)?, 161 variant are should be is? 434 “Proto-oncogene tyrosine-protein kinase receptor Ret has a role in cell growth and proliferation, differentiation” –syntax, 463 connector, and so on.
Author Response
Thank you very much for your comments and constructive observations.
- I’m wondering if the title is not too sophisticated, what is "second way or second direction" in the street?
Answer: Despite the low frequency of tumor-predisposing syndromes in patients with thyroid cancer, their identification is mandatory due to the associated risks for both the patient and the family. On the other hand, patients with these syndromes are at risk of developing thyroid cancer and require adequate follow-up in this regard.
- I’m not sure about the order of the syndromes; is it a frequency in Thyroid cancer or the frequency of the syndrome in the population?
Answer: The order of the syndromes in the table is established based on: the type of thyroid cancer, the frequency of thyroid cancer, the frequency of the disease, the overlapping phenotype.
- I think that both the introduction and conclusion could be more developed, especially the conclusion in the context of practical patients diagnostics… In the introduction, authors could add some more details about
Answer: We introduced the histologic classification of thyroid neoplasms released in 2022 by the World Health Organization in the introduction and some epidemiological information about thyroid cancer. We rephrased and added some elements of importance of practical diagnostics
- I’m wondering if Table 1 could have an additional column with frequency in thyroid cancer, maybe just percentage or "rare"?
Answer: We added a paragraph with the percent of overall thyroid cancer which correspond to some cancer predisposition syndromes
- Please check the correctness of all citation i.e 21, 82 (not cancer), 91 perhaps some citation should be split because they do correspond to the text and are confusing.
Answer: We modified according to your suggestions
- Please check the correctness of all citation i.e 21, 82 (not cancer), 91 perhaps some citation should be split because they do not correspond to the text and thus are confusing.
Answer: We modified according to your suggestions
- In line 473 problem with the citation format?
Answer: We introduced the correct reference
- A strong point that is crucial for my final recommendation is that I could not find any similar article published recently that would cover the same topic or idea.
Answer: Thank you for your appreciation.
1.
Please carefully check “the” and “a”, syntax, and gramma i.e. Lines: 101 others report (s)?, 161 variant are should be is? 434 “Proto-oncogene tyrosine-protein kinase receptor Ret has a role in cell growth and proliferation, differentiation” –syntax, 463 connector, and so on.
Answer: We made the grammatical corrections. In genetics is more used the term ‘‘Janus effect’’.
Round 2
Reviewer 1 Report
The manuscript has been significantly improved. I am satisfied with the Authors’ responses to my comments. I suggest that it is ready for publication.
Reviewer 3 Report
The manuscript has been sufficiently improved.
Minor issues:
1.Line 38 "is being"?
2.Line 64 "into" instead in
3. Line 546 earlier diagnosis? instead diagnostic